# Design and Detection of Cyanide Raman Tag pH-Responsive SERS Probes

**DOI:** 10.3390/bios13010021

**Published:** 2022-12-25

**Authors:** Jingjing Shen, Guan Liu, Wen Zhang, Wenwen Shi, Yang Zhou, Zejie Yu, Qunbo Mei, Lei Zhang, Wei Huang

**Affiliations:** 1Institute of Advanced Materials (IAM), Nanjing University of Posts and Telecommunications (NJUPT), 9 Wenyuan Road, Nanjing 210023, China; 2Frontiers Science Center for Flexible Electronics (FSCFE), Shaanxi Institute of Flexible Electronics (SIFE), Northwestern Polytechnical University (NPU), 127 West Youyi Road, Xi’an 710072, China; 3Frontiers Science Center for Flexible Electronics (FSCFE), Shaanxi Institute of Biomedical Materials and Engineering (SIBME), Northwestern Polytechnical University (NPU), 127 West Youyi Road, Xi’an 710072, China

**Keywords:** SERS probe, cyano Raman Tag, pH responsive, environment detection

## Abstract

As one of the most important parameters of biochemical analysis and detection, the pH value plays a very important role in cell function, food preservation and production, soil and water sources, and other applications. This makes it increasingly important to explore pH detection methods in depth. In this paper, a pH-responsive SERS probe based on the cyano Raman Tag was designed to realize pH sensing detection through the influence of the pH value of analytes on the displacement of the cyano Raman peak in the SERS probe. This cyano Raman tag exhibited not only excellent sensitivity in the liner range of pH 3.0–9.0 with a limit of detection (LOD) of pH 0.33, but also the anti-interference performance and stability (the relative standard deviation (RSD) was calculated to be 6.68%, *n* = 5). These results indicated that this pH SERS probe with the Raman cyano tag can provide new research ideas for future biological detection, bioimaging, and environmental detection.

## 1. Introduction

It is well-known that pH value is one of the important parameters in a variety of chemical reactions, biological processes, and life forms. For example, changes in the microenvironmental pH generated from survival, proliferation, apoptosis, and deterioration of living cells are used for cancer prediagnostics [1,2]. The detection of urine’s pH value can be used to diagnose kidney diseases [3,4,5], and the degree of damage to the ecosystem caused by environmental acidification can be judged by the detection of the pH value of lakes and rivers [6,7]. Currently, the main methods for detecting pH are ultraviolet–visible region (UV–Vis) spectroscopy [8,9], the colorimetric method [10,11], fluorescence spectroscopy [12,13], and potentiometry [14,15]. Based on the requirements of high sensitivity, fast response time, and no damage to the test object in pH detection, sensors using luminescence, colorimetric, and surface-enhanced Raman scattering (SERS) detection techniques are considered to be common equipment for pH detection [16,17,18].

As an optical sensing technology, SERS not only can directly obtain the fingerprint information of very low concentration molecules, but also qualitatively analyze a variety of analytes [19], as is often performed in the fields of biological imaging [20,21] and biochemical analysis [22,23]. SERS probes used target molecules to connect to the surface of metal nanomaterials to achieve Raman signal enhancement, which requires physical and chemical stability, high sensitivity, and multiplexing capability. At present, SERS probes mainly include temperature SERS probes [24,25], bioimaging SERS probes [26,27], and pH-responsive SERS probes [28,29]. However, pH-responsive SERS probes mainly used pH-sensitive molecules to functionalize SERS substrates and caused specific groups in the sensitive molecules to undergo protonation or deprotonation through the pH change of the surrounding environment, resulting in changes in the vibration peak intensity or peak displacement of some groups in the Raman spectrum in order to realize pH sensing. Based on the current studies on the pH SERS probe, it was found that, to achieve the pH response of the SERS probe, the selected pH-sensitive molecules should have the amino group, carboxyl group, and pyridine ring [30]. Representative pH-sensitive molecules in related studies included 4-aminobenthiol (4-ATP) [31,32,33], 2-aminobenthiol (2-ABT) [34], 4-mercaptobenzoic acid (4-MBA) [35,36,37,38], and 4-mercaptopyridine (4-MPy) [18,39].

Due to a large number of molecules in the fingerprint region (<1800 cm^−1^), the molecular Raman vibration peaks often overlap with the background signals, and it is difficult to distinguish the Raman vibration peaks of the same species in different environments [40]. Based on the unique characteristics of the Raman vibration peak of the alkyne (C≡C), cyano (C≡N), azide, deuterium, and metal−carbonyl groups in the Raman silent region (1800–2800 cm^−1^) [41], it has also been deeply studied in the design of related SERS probes [42,43,44,45,46]. In addition, sulfur-containing compounds (thiols, thioethers, and dithioethers) have a strong affinity for the Au surface, which can be used to form stable monocouches by Au-S polar covalent bonding [47]. Consequently, sulfur-containing compounds are often used in the surface modification of noble-metal nanomaterials and related studies on SERS probes’ design [35,48,49].

In this study, we designed a cyano Raman tag pH-responsive SERS probe. The benzyl 4-(9-(6-cyanopyridin)-9H-carbazole) 5-(1,2-dithiother) pentanoate (BCCDP) containing a pyridine ring, a disulfide bond, and a cyano group was selected as the pH-sensitive molecule. The selected pH-sensitive molecular formula is shown in Figure 1A. The pH-sensing detection was achieved by using the displacement of the cyano Raman vibration peak caused by the protonation/deprotonation effect of the N lone pair electrons on the pyridine ring due to the difference in the external pH value, as shown in Figure 1B,C. We prepared a pH-responsive SERS probe based on the cyano Raman tag, using the method of solvent evaporation induction (sol film formation) [50,51] and immersion self-assembly [52,53]. The SERS substrates prepared by gold nanoparticles (AuNPs) were dense and could generate a large number of “hot spots”, which showed good SERS response ability against the pH value. After the SERS probe was constructed with a 10^−8^ M pH-sensitive molecule, the Raman pH probe exhibited excellent pH liner responsibility and stability in buffer solution with a pH range from 3.0 to 9.0, and similar results were obtained for a real river sample with an RSD of 6.68% (*n* = 5). It was known that the pH-responsive SERS probe designed with cyano Raman tag was sensitive and reliable during detection.

## 2. Materials and Methods

### 2.1. Materials

Hydrogen tetrachloroaurate (III) trihydrate and sodium citrate tribasic dihydrate were purchased from Sigma-Aldrich (Shanghai, China). Sodium chloride (NaCl, 99.8%), potassium chloride (KCl, 99.8%), magnesium chloride hexahydrate (MgCl_2_·6H_2_O, 98.0%), cesium chloride (CsCl, 99.5%), cupric chloride (CuCl_2_, 99.0%), iron (III) chloride hexahydrate (FeCl_3_·6H_2_O, 99.0%), sodium hydroxide (NaOH, 96.0%), sodium nitrate (NaNO_3_, 99.0%), sodium iodide (NaI, 99.0%), sodium bromide (NaBr, 99.0%), sodium sulphate (Na_2_SO_4_, 99.0%), dibasic sodium phosphate (Na_2_HPO_4_, 98.0%), sodium hydroxide (H_3_PO_4_, 85.0%), boric acid (H_3_BO_3_, 99.8%), citric acid (C₆H₈O₇, 99.8%), acetic acid (CH_3_COOH, 99.8%), acetone (CH_3_COCH_3_, 99.0%), and ethanol (CH_3_CH_2_OH, 75.0%) were purchased from Sinopharm Chemical Reagent Co., Ltd. (Shanghai, China). BCCDP was purchased from Shanghai Medicilon Inc. (Shanghai, China). The above reagents were used directly without further purification. All aqueous solutions in our experiments were made with Milli-Q water (18 MΩ·cm^−1^).

*Preparation of Britton–Robinson (B–R) buffer solution:* First, 400 μL H_3_PO_4_, 240 μL CH_3_COOH, and 0.248 g H_3_BO_3_ were added to Milli-Q water and mixed to a constant volume of 100 mL. Second, the prepared mixed acid solution and 0.2 M NaOH solution were mixed into buffer solutions with different pH values at different volume ratios, and the pH value of the buffer solution was calibrated by a pH meter.

*Preparation of disodium hydrogen phosphate–citrate buffer solution:* First, 100 mL of 0.2 mol/L Na_2_HPO_4_ solution and 100 mL of 0.1 mol/L citric acid solution were prepared. Second, the prepared Na_2_HPO_4_ and citric acid solutions were mixed into buffer solutions with different pH values at different volume ratios, and the pH value of the buffer solution was calibrated by using a pH meter.

### 2.2. Test Instrumentation

Extinction spectra measurements of as-synthesized AuNPs were achieved by using a Shimadzu UV3600 spectrophotometer. Transmission electron microscopy (TEM, JEM-2010 instrument, Tokyo, Japan) was utilized to acquire the morphological information of as-synthesized AuNPs. The pH value of the solution was captured by Pondus Hydrogenii (PH meter, PHS-3CW, Shanghai, China).

Scanning electron microscope (SEM) images were acquired by using an S4800 instrument (Tokyo, Japan). The dark-filed microscopy (DFM) images and SERS spectrum measurements were collected by a −75 °C cooled CCD detector (PIXIS 400BR: excelon, Princeton Instruments) and an inverted microscope (eclipse Ti-U, Nikon) equipped with a monochromator (Acton SP2358). A stabilized 633 nm laser diode was used as the excitation light source. The microscope was equipped with a dark-field condenser (0.8 < numerical aperture (NA) < 0.95), a 60× objective lens, a 1200 grating with BLZ of 750 nm in the monochromator, and a true-color digital camera (Nikon DS-f2). The Raman sensitivity was calculated to be 0.816 cm^−1^.

### 2.3. Synthesis of AuNPs [54]

*Synthesis of Au seeds:* A solution of 22 mM sodium citrate (5 mL) in Milli-Q water (45 mL) was heated with an oil bath in a 100 mL two-necked round-bottomed flask for 15 min under vigorous stirring. At the same time, a condensing tube was used to prevent solvent evaporation. After boiling had commenced, 0.4 mL of HAuCl_4_ (25 mM) was injected. The color of the solution changed from yellow to bluish gray and then to soft pink in 10 min. The resulting Au seeds were coated with negatively charged citrate ions and thus could be well suspended in Milli-Q water.

*Seeded growth of AuNPs of up to 55 nm in diameter:* The synthesized Au seeds were immediately cooled in the reaction flask until the temperature of the temperature solution reached 90 °C. Then 1 mL of sodium citrate (60 mM) and 0.33 mL of a HAuCl_4_ solution (25 mM) were sequentially injected (time delay ∼of 3 min). After 30 min, the reaction was finished. This process was repeated twice. After that, the sample was diluted by extracting 25 mL of the sample and adding 24 mL of Milli-Q water. This solution was then used as seed solution, and the process was repeated. The concentration of each generation of NPs was approximately the same as the original seed particles. After about four rounds, the diameter of the AuNPs could reach 55 nm.

### 2.4. Preparation of SERS Substrates

In the experiment, we chose clean ITO glass slides as the platform of the SERS substrate. Firstly, the ITO glass slides were washed with dish soap, acetone, ethanol, and Milli-Q water and sequentially submitted to ultrasonication for 1 h. To obtain a clean SERS substrate platform, the ITO glass slides were washed again with Milli-Q water and dried with nitrogen.

To reduce the subsequent influence on SERS probe detection, it was necessary to remove the sodium citrate protectant from the AuNPs’ surface. Firstly, 1 mL of AuNPs solution was centrifuged three times at 4500 rpm for 5 min. Secondly, the AuNPs after the third centrifugation were enriched to 10 μL (5 nM) and subjected to ultrasonic treatment for 15 min. After considering the “hot spot” effect of SERS, combined with the wide application of the colloidal NPs film formation process caused by solvent evaporation [50,51], we prepared SERS substrates by solvent evaporation induction (sol-film formation).

### 2.5. Preparation of SERS Probe and Measurements

First of all, the BCCDP was dissolved in a certain volume of dichloromethane and ethanol solution containing 10^−3^, 10^−4^, 10^−5^, 10^−8^, and 10^−10^ M, respectively. Then the immersion self-assembly method [52,53] was selected for the fabrication of the SERS probe. As is typical, the prepared SERS substrate was immersed in five different concentrations of molecular BCCDP solutions, removed after a certain period of time, washed three times with Milli-Q water, and dried with nitrogen gas. Finally, the SERS spectra were recorded under the excitation wavelength of 633 nm lines. All of the spectra were recorded at a 29.2 μW laser power, spectrum collection time of 20 s, and five accumulations.

For SERS spectral data processing, Origin 8.5 software was used to subtract the background and fitting by Gauss function model to obtain the information of SERS peak intensity and position.

## 3. Results and Discussion

### 3.1. Characterization of the AuNPs

At present, most precious-metal nanomaterials were colloidal nanoparticle assemblies assembled by surfactants, and their surface protectants will produce unnecessary noise in SERS detection [55,56,57]. In addition, the intensity and position functions of plasmons excited by noble-metal nanoparticles produce SERS enhancement effects [58,59]. Hossain’s study found that AuNPs with a diameter of about 50 nm had the strongest enhancement ability in SERS [60].

To synthesize AuNPs with the easy removal of protective agent and about 50 nm, we synthesized AuNPs with sodium citrate system by using the seed-growth method [54]. In Figure 2(Aa–d), the AuNPs grown in seed, the first step, the third step, and the fifth step were characterized by TEM, respectively. We also performed size statistics separately in Figure 2(Ae–h), which could prove that AuNPs of 55 ± 5.1 nm were synthesized by Au species after five steps of growth. The optical properties of the Au seeds, as well as AuNPs during growth, were measured by UV–Vis spectra. As shown in Figure 2B, the data were normalized for the convenience of comparison, the locally amplified spectral map showed that AuNPs diameter increased with the growth of Au seed, and the absorption peak redshifted from 520 to 537 nm.

### 3.2. Characterization of pH Sensitive Molecule

As shown in Figure 3A, the 2240 cm^−1^ band corresponds to C≡N stretching vibration, the 1627 cm^−1^ band corresponds to C=N stretching vibration, the 1595 and 1615 cm^−1^ vibration bands correspond to C=C-N ring deformation antisymmetric stretching and symmetric stretching, and the 1500 and 1534 cm^−1^ vibration bands correspond to aromatic C=C stretching vibration. The 1468 cm^−1^ vibration band corresponds to the tensile vibration of C-N, the 1382 cm^−1^ vibration band corresponds to the coupling between the tensile vibration of the benzene ring and the tensile vibration of the carboxyl group, and the 1379 cm^−1^ vibration band corresponds to the stretching vibration caused by two different hydrogen bonds in the pyridine ring. The 1334 cm^−1^ vibration band corresponds to the deformation vibration of CH_2_, and the 1275 cm^−1^ vibration band corresponds to the tensile vibration of C-O.

To compare the Raman spectral information of BCCDP before and after assembly with SERS substrates, theoretical calculations were performed. The calculation was performed by using the Gaussian 09 suite of programs. The ground-state structure of the Au_4_-BCCDP molecule was optimized by using DFT with the B3LYP functional at the level of 6–31 g (d, p) for C, H, O, N, S, and Couty–Hall-modified LANL2DZ for Au atoms, respectively. A correction factor of 0.952 was adopted for all the computed wavenumbers corresponding to the vibrational modes. It could be seen in Figure 3B that the theoretical calculation of the molecule was consistent with the actual measurement. After the molecule was assembled with the SERS substrate, the Raman summit of the cyanide group was blue-shifted from 2244 to 2164 cm^−1^.

### 3.3. Characterization of SERS Probe

To verify whether pH-sensitive molecules self-assemble with the SERS substrate and to explore the detection sensitivity of the SERS probe, we selected pH-sensitive molecule solutions with 10^−3^, 10^−4^, 10^−5^, 10^−8^, and 10^−10^ M concentrations to self-assemble with the SERS substrate to construct the SERS probe. Figure 4 shows the Raman spectra of the SERS probe constructed at the above five molecular concentrations. It can be seen from the Raman spectra that there were more groups in the fingerprint region of the molecule and that the vibration peak signals around 1600 cm^−1^ overlap greatly. In the Raman silent zone, it was found that two vibration peaks (2150 and 2240 cm^−1^) appeared around 2200 cm^−1^ when the concentration of the BCCDP solution was more than 10^−5^ M. Combined with DFT theoretical calculation and analysis, it could be inferred that the stretching vibration peak at 2240 cm^−1^ was caused by the free state of the pH-sensitive molecule, while the vibration peak at about 2150 cm^−1^ coincides with the vibration peak at 2164 cm^−1^ of Au_4_-BCCDP. In addition, we also performed Raman spectroscopy on the SERS probe constructed with pH-sensitive molecules at concentrations below 10^−10^ M, but no distinct Raman spectral information was detected.

Due to the coffee-ring effect brought on by the solvent evaporation induction (sol-film forming) method, the uniformity and repeatability of the SERS substrate prepared would be affected. DFM, SEM, and SERS spectroscopy were employed to evaluate the homogeneity of the SERS probe constructed with BCCDP at a concentration of 10^−8^ M. As shown in Figure 5A, it could be seen that the dark-field image of the SERS substrate was purplish red, which indicated that the compact aggregation of AuNPs lead to the notable red-shift in the localized surface plasmon scattering spectra. Moreover, it could be further proved in the in−situ SEM image (Figure 5B). As shown in Figure 5C, four Raman spectra were randomly selected on different positions of the SERS substrate to analyze the peak intensity and peak shift of the cyano vibration peak, respectively. The RSD of the cyano vibration peak was calculated to be 0.03% of peak position and 6.62% of peak intensity (*n* = 5), which indicated the better uniformity of the SERS probe compared with the probe constructed with 10^−10^ M BCCDP solution.

We considered that the strong laser power would lead to the desorption and photodegradation of the SERS tag molecules in the process of the SERS measurement, so we explored the effect on SERS probes at different laser powers. As shown in Figure 6, different laser powers with 10, 29, and 40 μW, were used to continuously monitor the changes of the SERS spectra. It was found that the SERS signal intensity was weak and the signal–noise ratio was lower than 3 with the 10 μW laser power. Moreover, the peak intensity of cyano Raman peak decreased gradually with the increase of measurement time when the laser power was 40 μW. Above all, the SERS measured in our subsequent experiments were carried out by using 29 μW laser power with the SERS substrates constructed with 10^−8^ M BCCDP.

To verify the stability of the SERS probe, we performed interference experiments under the conditions of anions and cations, acid–base cycles, and different types of buffer solutions. As shown in Figure 7A,B, we explored the effects of cationic solutions (Na^+^, K^+^, Cs^+^, Fe^3+^, Mg^2+^, and Cu^2+^) and anions (Cl^−^, Br^−^, I^−^, and SO_4_^2−^) at a concentration of 10^−5^ M on the stability of the SERS probe. Figure 7(Ab,Bb) show that the negative and positive ions had little effect on the displacement of the cyano vibration peak of the SERS probe, which maintained a relatively stable state. Figure 7C,D show the interference inquiry experiment of the SERS probe with acid–base cycle (pH = 3 and pH = 11.28) under disodium phosphate–citric acid and B–R buffer. It could be seen from Figure 7(Cb,Db) that, under the two types of buffer solutions, the cyano Raman vibration peak in the SERS probe showed a redshift under acidic conditions compared with alkaline conditions and presented a reciprocating and cyclic change.

To test the pH-response ability of the SERS probe, the SERS spectra were measured in the range from 3.0 to 11.28 in disodium hydrogen phosphate–citric acid buffer solution. Figure 8 explores the changes in the response ability of the cyano Raman vibration peak of the SERS probe in the range of pH = 3, 4, 5, 6, 7, 8, 9, 10.39, and 11.28. It was found that the continued linear blue-shift of the cyano Raman peak in the range of pH 3.0−9.0, the fitting line was y = 2152.00 − 2.10x, the R^2^ was calculated to be 0.9784, and the limit of detection was calculated to be pH 0.33, with an S/N of 3.

## 4. Detection Application of SERS Probe

Based on the current aggravation of environmental pollution, the detection of the pH value of water resources has become particularly important [61]. Therefore, we selected river water for our real sample analysis to verify the detection performance and feasibility of the cyano Raman tag pH SERS probe. The river water was collected from a river in Nanjing, Jiangsu Province, and the suspended solids of the river water were removed by centrifugation.

Figure 9 shows the Raman detection of five selected groups of river water on the cyano Raman tag SERS probe constructed with BCCDP at the concentration of 10^−8^ M, and the pH value corresponding to the displacement of the cyano Raman vibration peak after testing was calculated and analyzed by using the function model in Figure 8B. The detection results of five groups of samples are shown in Figure 9B. Compared with the pH values measured by the pH meter and the pH values calculated by the model after the SERS probe detection of river water samples in Table 1, the analysis showed that the average pH measured by the pH meter is 7.23, and the average pH calculated by the SERS probe is 7.71, and the RSD was 6.68% (*n* = 5).

In addition, different proportions of disodium hydrogen phosphate and citric acid were added to the river-water samples to change the pH value of the river water (pH = 3, 6.1, 7.4, and 9.1) in order to further test the SERS probe’s performance, and the test result is shown in Figure 10B. Compared with the pH value measured by the pH meter and the pH value calculated by the function model after SERS probe detection in Table 2, the RSD of the pH-value-detection results was 3.90% (*n* = 5).

In general, in regard to the detection and analysis of the river-water sample, this pH SERS probe constructed based on cyano Raman tag had a good performance.

## 5. Conclusions

In this work, a pH-responsive SERS probe based on the cyano Raman tag was successfully designed and prepared by immobilization of BCCDP on the surface of 55 nm AuNPs. After immersion in 10^−8^ M BCCDP solution for a certain time, this probe exhibited the best response ability to pH value at 29 μW laser power with 20 s collection time. Moreover, the cyano vibration peak of this SERS probe would shift due to the protonation/deprotonation effect of N in the pyridine ring under different acid–base conditions. Thus, this probe could be used to evaluate the level of acid or alkali of buffer solution and river-water samples. As expected, this SERS probe showed a liner relationship between the Raman shift of the C≡N group and the pH value. The LOD of this probe reached pH 0.33, and the RSD of detection in the buffer solution and real river-water sample was 6.68% and 3.90%, respectively. Therefore, this study is helpful for future exploration of environmental water-resource detection, food preservation and production, and soil acid–base detection. Furthermore, since the pH detection range of cancer cells is between 4 and 7 [62], this SERS probe with a cyano Raman tag would provide an alternative selection for the detection of the pH value or pH imaging in cancer cells due to the fact that its typical peak located at 2150 cm^−1^ supplied a high signal–noise ratio in the biologically silent region.

## Figures and Tables

**Figure 1 biosensors-13-00021-f001:**
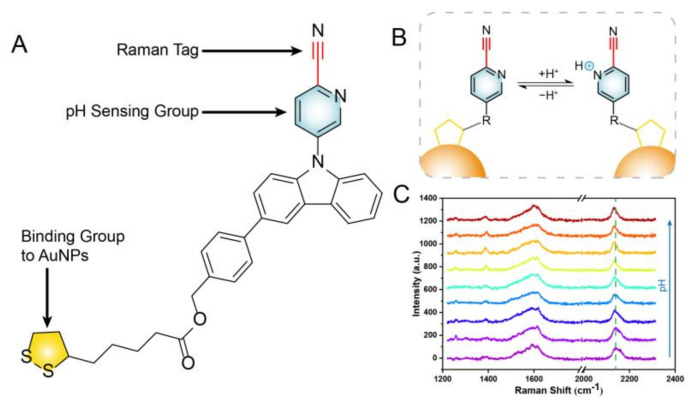
(**A**) The formula of the pH−sensitive molecule BCCDP. (**B**,**C**) Schematic of the sensing detection mechanism of cyano Raman tag pH−responsive SERS probe.

**Figure 2 biosensors-13-00021-f002:**
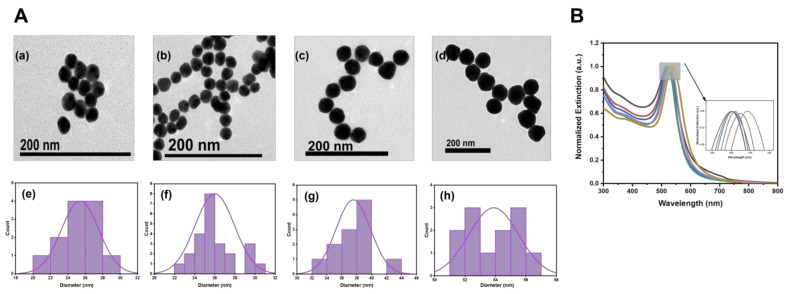
(**A**) TEM images (**a**–**d**) and corresponding grain-size statistics (**e**–**h**) of seeds and first-step, third-step, and fifth-step AuNPs. (**B**) UV−Vis absorption spectra of Au seeds and AuNPs during growth: Au seed (black line), growth 1 step (red line), growth 2 step (blue line), growth 3 step (green line), growth 4 step (purple line), and growth 5 step (yellow line).

**Figure 3 biosensors-13-00021-f003:**
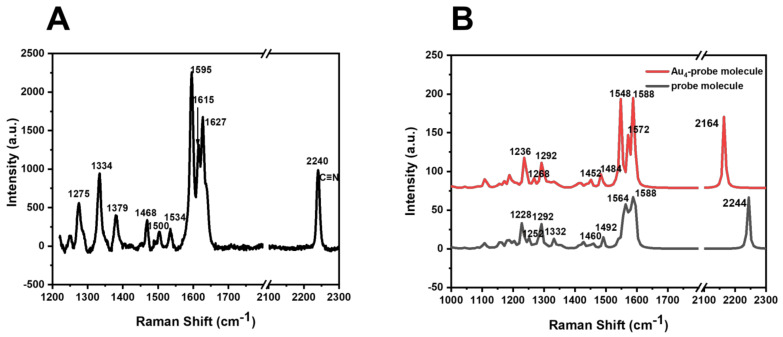
(**A**) Raman spectra of the pH-sensitive molecule of BCCDP. (**B**) DFT theoretical calculation of Raman spectra of pH-sensitive molecule and Au_4_-BCCDP.

**Figure 4 biosensors-13-00021-f004:**
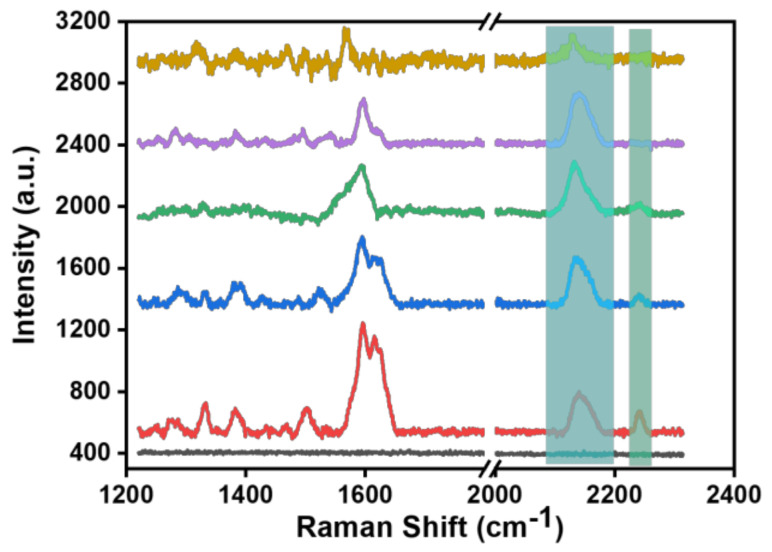
Raman spectra of the SERS probe constructed at 5 different concentrations of BCCDP: blank (black line), 10^−3^ M (red line), 10^−4^ M (blue line), 10^−5^ M (green line), 10^−8^ M (purple line), and 10^−10^ M (brown line).

**Figure 5 biosensors-13-00021-f005:**
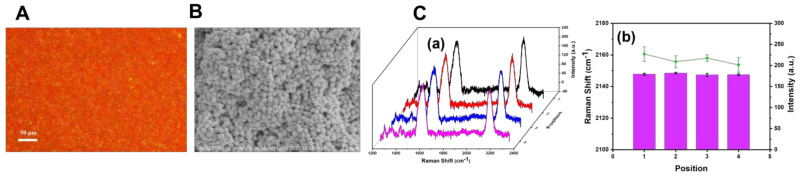
(**A**) Dark-field image of SERS substrates. (**B**) Scattering images of SERS substrates taken by SEM. (**C**) Raman spectra of four randomly selected data points on the SERS substrate (**a**) and the analysis results of the peak intensity and displacement of the cyano vibration peak (**b**).

**Figure 6 biosensors-13-00021-f006:**
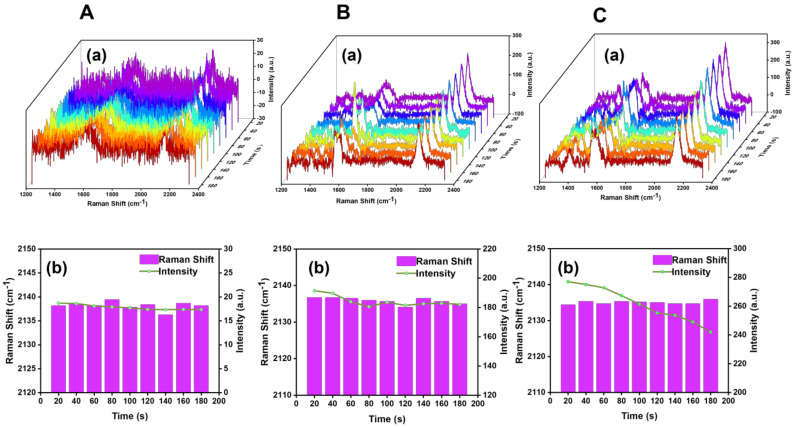
(**A**–**C**) Exploration of SERS probe spectra as a function of time at 10, 29, and 40 μW laser power, respectively. (**a**) Corresponding to the collected SERS probe Raman spectra. (**b**) Corresponding to the analysis results of the peak intensity and peak shift of cyano vibration peak.

**Figure 7 biosensors-13-00021-f007:**
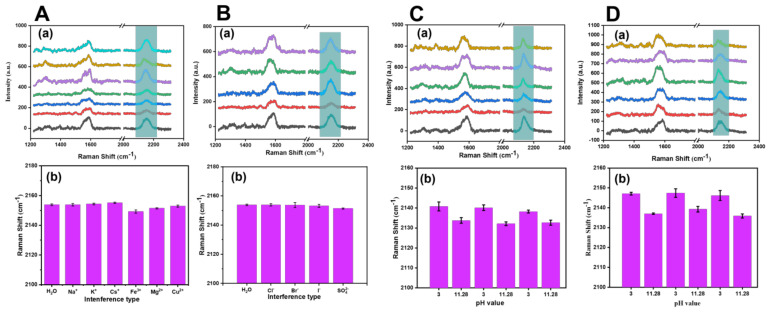
(**A**) Exploration of the influence of 10 ^−5^ M cationic solution: H_2_O (black line), Na^+^ (red line), K^+^ (blue line), Cs^+^ (green line), Fe^3+^ (purple line), Mg^2+^ (orange line), and Cu^2+^ (cyan line). (**B**) Exploration of the influence of 10 ^−5^ M anions solution, namely H_2_O (black line), Cl^−^ (red line), Br^−^ (blue line), I^−^ (green line), and SO_4_^2−^ (purple line), on the stability of the SERS probe. Exploration of the influence on the stability of the SERS probe under the interference of the acid-base cycle (pH = 3 and pH = 11.28) of disodium phosphate–citric acid buffer (**C**) and B–R buffer (**D**). (**a**) Corresponding to the collected SERS probe Raman spectra. (**b**) Corresponding to the analysis results of the peak displacement of cyano vibration peak.

**Figure 8 biosensors-13-00021-f008:**
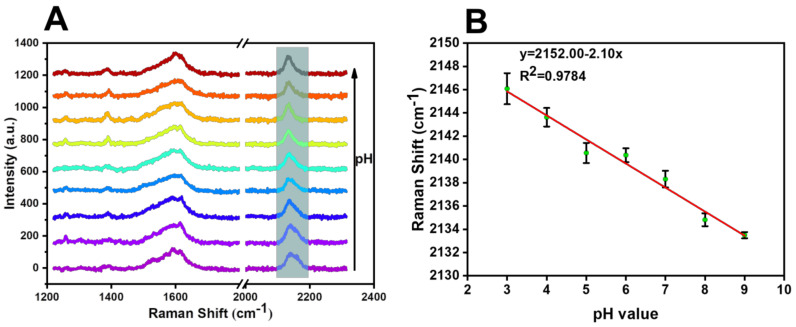
(**A**) Raman spectra of the response change of cyano Raman vibration peak of SERS probes in the range of pH = 3, 4, 5, 6, 7, 8, 9, 10.39, and 11.28. (**B**) Response fitting curve of cyano Raman peak displacement of SERS probes in the range of pH = 3−9.

**Figure 9 biosensors-13-00021-f009:**
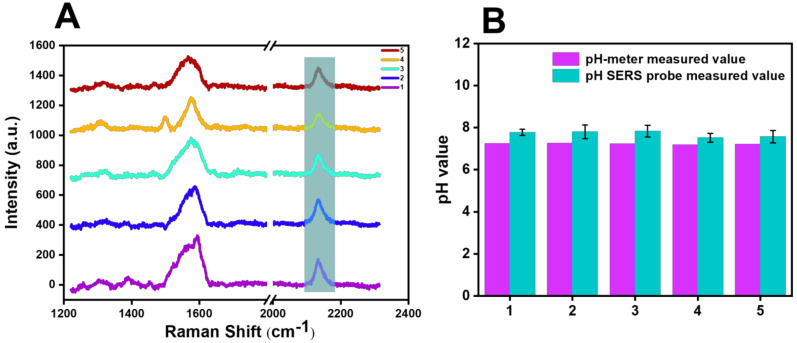
(**A**) Raman spectra of the 5 selected sets of river-water samples examined with the SERS probe. (**B**) Comparison of the results of pH values calculated after SERS probe detection and pH measured by pH meter for 5 groups of river-water samples.

**Figure 10 biosensors-13-00021-f010:**
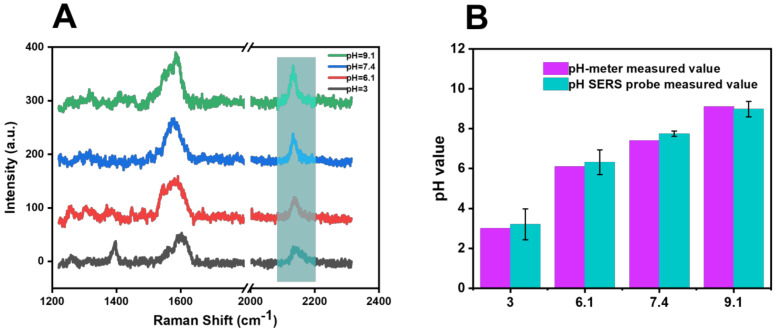
(**A**) Raman spectra of river −water samples with altered acid −base detected by SERS probe. (**B**) Comparison of the calculated pH results of river-water samples with altered acid −base detected by SERS probe and with pH results determined by pH meter.

**Table 1 biosensors-13-00021-t001:** Detection data of 5 groups of river-water samples.

Number	Raman Shift (cm^−1^)	pH-Meter Measured Value	pH SERS Probe Measured Value	RSD (100%)
1 2 3 4 5	2135.61 2135.61 2135.53 2136.19 2136.09	7.24 7.26 7.23 7.19 7.22	7.80 7.80 7.84 7.52 7.57	7.80% 7.50% 8.47% 4.71% 4.93%

**Table 2 biosensors-13-00021-t002:** Detection data of river-water samples with altered acidity and alkalinity.

pH-Meter Measured Value	Cyano Raman Shift (cm^−1^)	pH SERS Probe Measured Value	RSD (100%)
3 6.1 7.4 9.1	2145.27 2138.72 2135.73 2133.00	3.2 6.32 7.74 9.04	6.82% 3.67% 4.59% 0.53%

## Data Availability

Not applicable.

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
