# Peer review of "Design and Detection of Cyanide Raman Tag pH-Responsive SERS Probes"

_biosensors, 2022, doi:10.3390/bios13010021_

Round 1
Reviewer 1 Report
In this manuscript, the authors report the design of cyanide Raman tag pH responsive SERS probe and basing on the change of cyano Raman peak displacement, this probe was used to verify the acid-base performance of environmental river water samples. Although the SERS probe designed by the authors is relatively novel, the performance of SERS probe is proved to be good by using analytes, there are some doubts in the manuscript, so I recommend it accepted after revision. The details are listed below:
1. In the chapter where the author mentioned the preparation of SERS substrates, it was found that there was an error in the "10 mL" unit at Lines 151-152, and the author was advised to confirm it carefully.
2. By reviewing the authors' statements in the characterization section of AuNPs, it is recommended to cite supporting research descriptions at Lines 166-169 for more persuasive purposes.
3. After reviewing the chapter on pH sensitive molecule (BCCDP) mentioned by the author, it is suggested that the representation of group Raman vibration peaks at Lines 189-199 should be shown in Figure 3 (A), which will make it more clear to the reader.
4. In the characterization of SERS probe, did the author consider that the peak intensity of cyano Raman peak in the designed SERS probe was related to the concentration of BCCDP molecule?
5. By reviewing Figure 6B(a) and (b), it is found that the author added blank control in the experiment, and it is suggested that the description should be added in the text.
6. In the investigation of pH response of SERS probe, why the author selected the Logarithm function Log3P1 distribution function model instead of linear fitting model for data processing in Figure 8(b).
Reviewer 2 Report
The manuscript reports on the realization of a SERS substrate for the detection of pH based on the conjugation of gold nanoparticles (AuNPs) with a cyano Raman probe. Authors evaluated the performances of the sensor with respect to the concentration of the Raman probe, acquisition time and interference of other ions.
The manuscript presents several criticisms that do not make it suitable for publication in the present form. A list of the major points is reported below:
1. lines 79-80: the authors state that the limit of detection (LOD) of the SERS substrate is 10-10 M. How was the LOD evaluated?
2. More details should be provided with respect to the preparation of the SERS probe and SERS measurements:
a) In which solvent was the SERS probe BCCDP dissolved?
b) Specify the magnification and the numerical aperture of the objective employed for SERS measurements.
c) Which is the spectral resolution of the acquired spectra? Which grating was employed?
3. Could authors identify other spectral markers witnessing the formation of covalent bond between the SERS probe and the AuNPs (e.g the formation of S-Au bond)?
4. How were the peak position and intensity evaluated? Did authors perform peak fitting? If so, please specify in the manuscript how the spectral analysis was performed.
5. Were the SERS measurements performed in liquid or were them performed after evaporation of the sample solution?
6. Concerning the measurements as a function of time reported in Figure 6, could the observed intensity variations be ascribed to plasmon-induced chemical modifications of the SERS probe? SERS measurement as a function of the laser power should be performed in order to evaluate possible degradation of the molecule.
7. One of the most critical aspects of the work concerns the application presented by authors. What is the advantage of employing a nanosensor for measuring the pH of bulk water, since the very same measurement can be obtained by a conventional pH-meter?
The advantage of a SERS-based pH sensor should reside in the possibility to evaluate the pH on the nanoscale, otherwise I don’t see the benefits of realizing a nanosensor to measure the pH of water.
Authors should better discuss this point and provide a realistic perspective for the application of the proposed SERS sensor.
Minor points:
1. Please, add a scalebar to the image of Figure 5A.
2. The label of the graphs are hardly readable, particularly in Figure 6, please enlarge the fonts.
3. SEM images of the SERS substrate should be provided.
Round 2
Reviewer 2 Report
Authors replied satisfactorily to the raised issues. The manuscript can be considered for publication.